# SegTalker: Segmentation-based Talking Face Generation with Mask-guided Local Editing

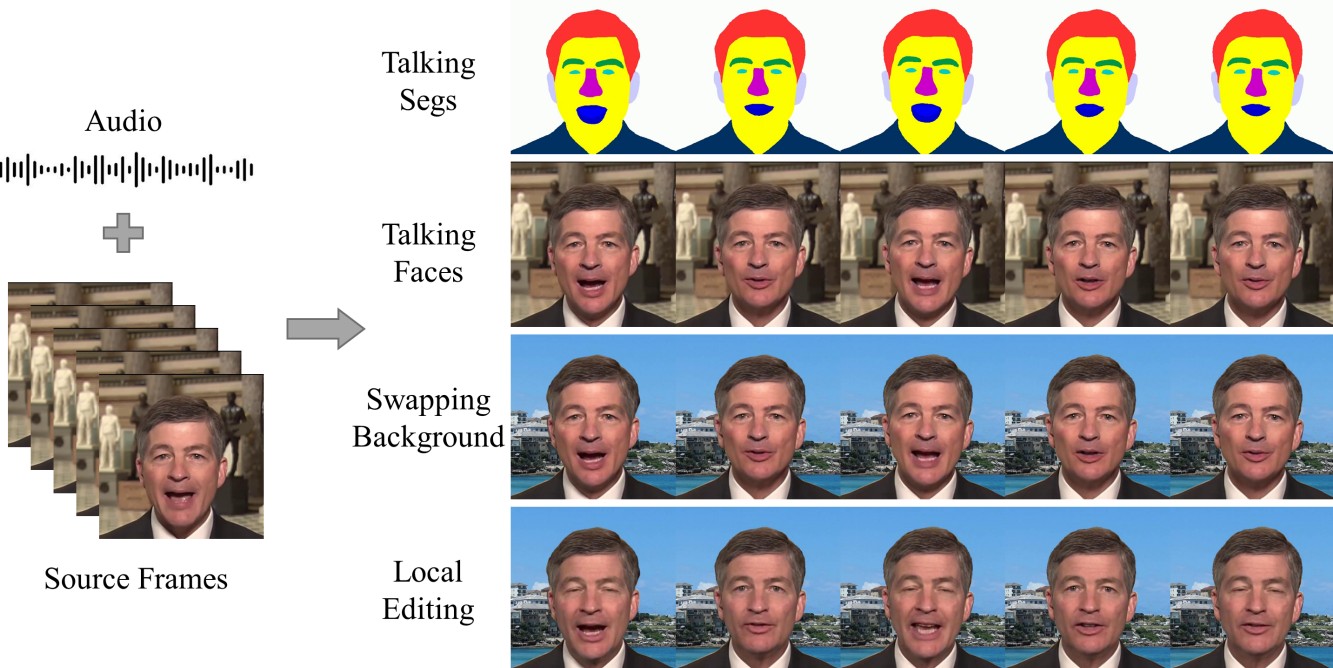

**Figure 1: Given a talking video and another speech, SegTalker can produce high-fidelity and synchronized video with rich textures (row 2), enabling swapping background (row 3) and local editing such as blinking (row 4).**

## ABSTRACT

Audio-driven talking face generation aims to synthesize video with lip movements synchronized to input audio. However, current generative techniques face challenges in preserving intricate regional textures (skin, teeth). To address the aforementioned challenges, we propose a novel framework called **SegTalker** to decouple lip movements and image textures by introducing segmentation as intermediate representation. Specifically, given the mask of image employed by a parsing network, we first leverage the speech to drive the mask and generate talking segmentation. Then we disentangle semantic regions of image into style codes using a mask-guided encoder. Ultimately, we inject the previously generated talking segmentation and style codes into a mask-guided StyleGAN to synthesize video frame. In this way, most of textures are fully preserved.

Moreover, our approach can inherently achieve background separation and facilitate mask-guided facial local editing. In particular, by editing the mask and swapping the region textures from a given reference image (e.g. hair, lip, eyebrows), our approach enables facial editing seamlessly when generating talking face video. Experiments demonstrate that our proposed approach can effectively preserve texture details and generate temporally consistent video while remaining competitive in lip synchronization. Quantitative results on the HDTF dataset illustrate the superior performance of our method over existing methods on most metrics.

## CCS CONCEPTS

• **Computing methodologies** → **Computer vision tasks**; • **Information systems** → **Multimedia content creation**.

## KEYWORDS

Video Generation, Talking Face Generation, Attribute Editing

## 1 INTRODUCTION

Talking face generation, which aims to synthesize facial imagery precisely synchronized with input speech, has garnered substantial research attention for its numerous applications in the fields of digital human, virtual conference and video dubbing [20, 42, 44, 52].

*MM '24, October 28-November 1, 2024, Melbourne, AUS*
© 2018 Copyright held by the owner/author(s). Publication rights licensed to ACM.
ACM ISBN 978-1-4503-XXXX-X/18/06. . . $15.00
https://doi.org/XXXXXXX.XXXXXXX

There are many attempts to realize high-fidelity talking face. Early approaches first predict mouth shapes from speech using recurrent neural networks, then generate the face conditioned on the shapes [37]. Recent end-to-end methods directly map speech spectrograms to video frames leveraging different intermediate representations [20, 36, 53, 55]. Zhang et al. [53] takes advantage of 3D Morphable Models (3DMMs), a parametric model that decomposes expression, pose, and identity, to transfer facial motions. Zhou et al. [55] employs the landmark as the representation. Meshry et al. [20] factorizes the talking-head synthesis process into spatial and style components through the use of coarse-grained masks, but they do not facilitate texture disentanglement and facial editing. More recently, Kicanaoglu et al. [14] performs unsupervised vector quantization on intermediate feature maps of StyleGAN to generate abundant semantic regions for local editing. Despite improvements in photo-realism, current talking face methods still face challenges in preserving identity-specific details such as hair, skin textures and teeth. Furthermore, within the current landscape of talking face generation methods, there is no single technique that can concurrently accomplish facial editing and background replacement. Our method elegantly incorporates facial editing into talking face generation in an end-to-end manner through the intermediate representation of segmentation.

In this paper, we aim to design a unified approach that realizes the controllable talking face synthesis and editing. We propose a novel framework termed **SegTalker** that explicitly disentangles textural details with lip movements by utilizing segmentation. Our framework consists of an audio-driven talking segmentation generation (**TSG**) module, followed by a segmentation-guided GAN injection (**SGI**) network to synthesize animation video. We utilize a pre-trained network [49] to extract segmentation as prior information to decompose semantic regions and enhance textural details, while also seamlessly enabling fine-grained facial local editing and background replacement. Specifically, given the input image and speech, we first conduct face parsing to obtain the segmentation. Subsequently, **TSG** module extracts image and speech embedding, then combines these embeddings to synthesize new segmentation with lips synchronized to the input speech. After that, **SGI** module employs a multi-scale encoder to project the input face into the latent space of StyleGAN [13]. Each facial region has a set of style codes for different layers of the StyleGAN generator. Then We inject the synthesized mask and style codes into the mask-guided generator to obtain the talking face. In this way, the structural information and textures of facial components are fully disentangled. Furthermore, facial local editing can be accomplished by simply modifying the synthesized mask or swapping the region textures from a given reference image, achieving seamless integration with talking face synthesis. Experiments demonstrate that our model synthesizes high-fidelity talking faces in visual quality and texture preservation with enhanced editing flexibility compared to existing state-of-the-art methods. In summary, our contributions are:

- **We propose a novel framework that utilizes segmentation as intermediate representation to disentangle the lip movements with image reconstruction for talking face generation, achieving consistent lip movements and preserving fine-grained textures.**

- **We employ a multi-scale encoder and mask-guided generator to realize the local control for different semantic regions. By manipulating the masks and smoothly swapping the textures, we can seamlessly integrate the facial local editing into the talking face pipeline and conduct swapping background.**

- **Experiments on HDTF dataset demonstrate our superiority over state-of-the-art methods in visual quality, id preservation and temporal consistency.**

## 2 RELATED WORK

### 2.1 Audio-driven Talking Face Generation

Talking face generation, which aims to synthesize photo-realistic video of a talking person giving speech as input, has garnered increasing research attention in recent years. With the emergence of generative adversarial networks (GANs) [9], many methods [20, 24, 36, 53, 55] have been proposed for synthesizing animation video. In terms of the intermediate representations, the existing works can be categorized into landmark-based, 3D-based and others. In the landmark-based methods, Suwajanakorn et al. [37] use recurrent neural network (RNN) to build the mapping from the input speech to mouth landmark, and then generate mouth texture. Zhou et al. [55] combines LSTM and self-attention to predict the locations of landmarks. Zhong et al. [54] utilizes transformer to predict landmarks, then combines multi-source features (prior information, landmarks, speech) to synthesize talking face. Recently, DiffTalk [31] takes speech and landmarks as conditioned inputs and utilizes a latent diffusion model [27] to generate talking faces. For 3DMM-based method, SadTalker [52] learns realistic 3D motion coefficients for stylized audio-driven single image talking face animation, achieving high-quality results by explicitly modeling audio-motion connections. Some styleGAN-based method such as StyleHEAT [48] leverages a pre-trained StyleGAN to achieve high-resolution editable talking face generation from a single portrait image, allowing disentangled control via audio. More recently, the emergence of neural radiance field (NeRF) provides a new perspective for 3D-aware talking face generation [10, 30]. However, these intermediate representations have difficulty in capturing fine-grained details and preserving identity i.e., teeth and skin textures which degrade the visual quality heavily. Wav2Lip [24] adopts the encoder-decoder architecture to synthesize animation videos. However, there are conspicuous artifacts with a low resolution in the synthesized videos. In this work, we employ a novel representation, segmentation, to disentangle lip movement with image reconstruction, and further extract per-region features to preserve texture details.

### 2.2 GAN Inversion

GAN inversion aims to invert real images into the latent space of pre-trained generator for reconstruction and editing. Several StyleGAN inversion methods have been proposed, they can typically be divided into three major groups of methods: 1) gradient-based optimization of the latent code [1, 2, 12, 29], 2) encoder-based [3, 25, 38, 46, 47] and 3) fine-tune methods [4, 21, 26, 50]. The gradient-based optimization methods directly optimize the latent code using gradient from the loss between the real image and the

generated one. The encoder-based methods train an encoder network over a large number of samples to directly map the RGB image to latent code. The gradient-based optimization methods always give better performance while the encoder-based cost less time. The fine-tuning methods make a trade-off between the above two and use the inverted latent code from encoder as the initialization code to further optimization. However, existing works focus on global editing and cannot make fine-grained control of the local regions. Our method uses a variation of [25] to realize local editing via manipulating a novel $\mathcal{W}^{c+}$ [19] latent space.

## 2.3 Mask-guided Facial Editing

Deep semantic-level face editing has been studied for a few years. Many works of StyleGAN priors have shown a semantic disentanglement property in spatial dimensions [14, 15, 20, 35]. Lee et al. [16] learn a mapping between semantic masks and images. Shi et al. [35] and Kim et al. [15] achieve more fine-grained editing via explicit semantic style injection and masks to factorize semantic regions. Recently, Kicanaoglu et al. [14] performs unsupervised clustering on StyleGAN's intermediate output features to acquire spatial semantics. Then an image-to-image (I2I) network [22] is employed to take the mask as conditional input and generate the edited image. However, such an approach is time-consuming, and the performance is heavily constrained by the efficacy of clustering. Following [35], We leverage mask labels as prior information to extract spatially semantic region features, achieving semantic disentanglement.

## 3 PROPOSED METHODS

### 3.1 Overview

To tackle with the lack of regional textures in talking face generation, we explicitly disentangle semantic regions by introducing segmentation mechanism. Leveraging segmentation as an intermediate representation, our approach decouples audio-driven mouth animation and image texture injection. The speech is solely responsible for driving the lip contours, while the injection module focuses on extracting per-region textures to generate the animation video. The overall framework of our proposed model, termed as **SegTalker**, is illustrated in fig. 2. The pipeline consists of two sub-networks: (1) talking segmentation generation (**TSG**) and (2) segmentation-guided GAN injection network (**SGI**), which are elaborated in section 3.2 and section 3.3, respectively.

### 3.2 Talking Segmentation Generation

The first proxy sub-network is the talking segmentation generation (TSG) module. Given speech and image frame, this network first employs parsing network [49] to extract mask, then synthesizes talking segmentation. The original network generates 19 categories in total. For the sake of simplicity, we merge the same semantic class (e.g. left and right eyes), resulting in 12 final classes. During pre-processing, video is unified to 25fps with speech sampled at 16kHz. To incorporate temporal information, Following [41], the global and local features are extracted as speech embedding. We employ **mask encoder** to extract visual embedding from two masks: a pose source and an identity reference. The two masks are concatenated in the channel dimension. The pose source aligns with the

target segmentation but with the lower half occluded. The identity reference provides facial structural information of the lower half to facilitate training and convergence. Without concerning textural information, the model only focuses on learning the structural mapping from speech to lip movements.

We employ a CNN-based network to extract embedding of a 0.2-second audio segment whose centre is synchronized with the pose source. Similar to text, speech always contains sequential information. To better capture temporally relevant features, we employ the pre-trained AV-Hubert [33, 34] as a part of **audio encoder** to extract long-range dependencies. AV-Hubert has conducted pre-training for audio-visual alignment, so the extracted embedding is very close to the semantic space of the video. When using AV-Hubert to extract audio embedding, we only need to feed speech and the visual signal is masked. Specifically, given a 3s speech chunk, we feed it into the Transformer-based AV-Hubert to produce contextualized speech features. We then extract the feature embedding corresponding to the given image segment. Given the mixed speech embedding and visual embedding, **Generator** synthesizes the final talking segmentation. We adopt U-Net [28] as the backbone architecture. In addition, skip connections and transposed convolutions are utilized for feature fusion and up-sampling.

Given an mask synthesized by the model and the ground truth mask, we employ two types of losses to improve generation quality i.e., the reconstruction loss and the syncnet loss.

**Reconstruction Loss** Unlike previous generative tasks that often synthesize RGB image and adopt L1 loss for reconstruction, talking segmentation involves generating segmentation where each pixel denotes a particular class. To stay consistent with semantic segmentation task, we employ cross entropy loss as our reconstruction loss. Given $N_i$ generated masks $y_i$ and ground truth $\hat{y}_i$ with a categories of $M$ regions, the cross entropy loss is defined as:

$$\mathcal{L}_{ce} = \frac{1}{N_i} \sum_{i}^{N_i} \sum_{c=1}^{M} y_{ic} \log\left(\hat{y_{ic}}\right) \tag{1}$$

where $y_{ic}$ denotes the one-hot encoded vector for the i-th generated segmentation belonging to the c-th category. For generated segmentation, different classes occupy varying proportions of areas. Semantically important regions like lips and eyes constitute small fractions, while background dominates most areas. To mitigate the class imbalance issue, a weighted cross entropy is formulated as:

$$\mathcal{L}_{w\text{-}ce} = \frac{1}{N_i} \sum_{i}^{N_i} \sum_{c=1}^{M} w_c \, y_{ic} \log\left(\hat{y_{ic}}\right) \tag{2}$$

where $w_c$ denotes the weight for the corresponding category and is determined on the inverse proportionality of the areas of different regions on the whole dataset.

**SyncNet Loss** The reconstruction loss mainly restores images at the pixel level without effective semantic supervision. Therefore, we train a segmentation-domain SyncNet from [24] to supervise lip synchronization. During training, a speech chunk is randomly sampled from speech sequences, which can be either synchronized (positive example) or unsynchronized (negative example). The SyncNet consists of a speech encoder and a mask encoder. For the mask, we use one-hot encoding as input and concatenate $T_v$ masks along the channel dimension. Specifically, the SyncNet takes inputs of a

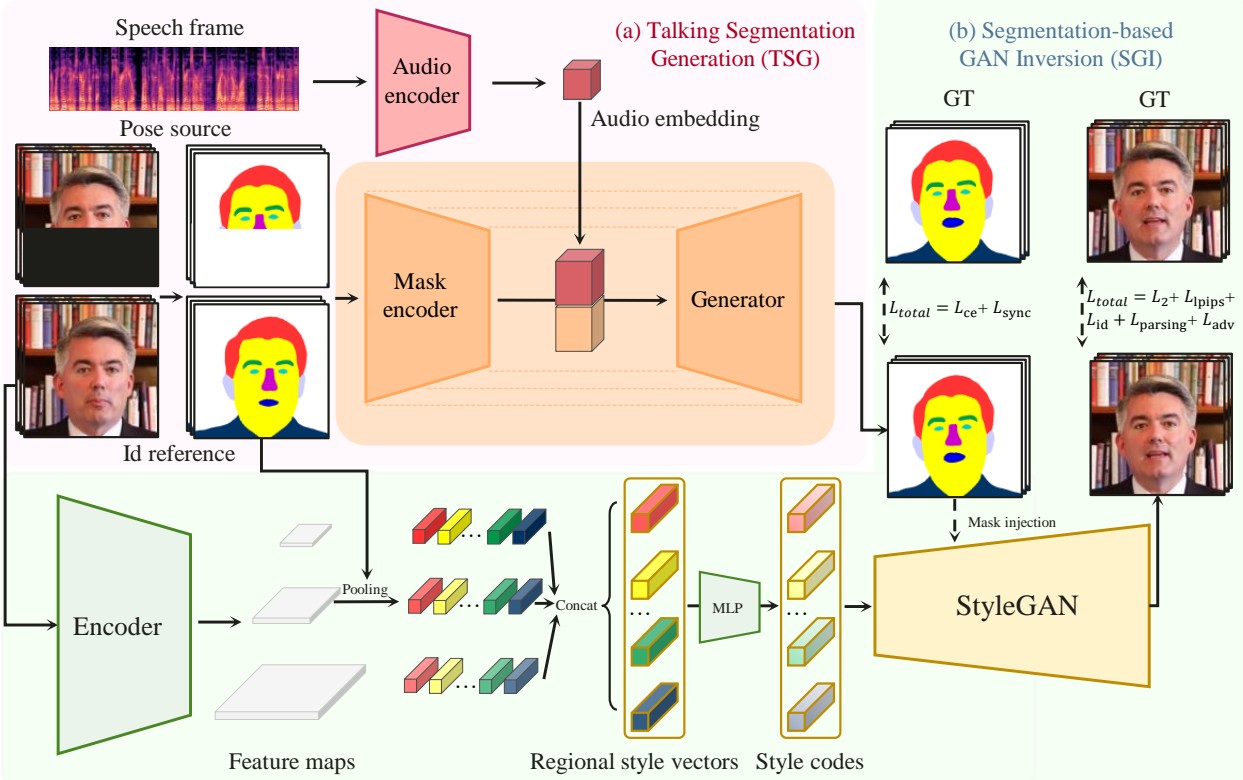

**Figure 2: Overview of the proposed SegTalker framework for talking face generation. (a) talking segmentation generation (TSG) module takes mel and mask as inputs, then synthesizes the talking segmentation with lip synchronized to input speech. (b) Given reference image and mask from TSG, segmentation-guided GAN injection (SGI) network utilizes a mask-guided multi-scale encoder to extract different semantic region codes, then injects the style codes and synthesized mask from TSG into the mask-guided generator to obtain the final talking face image.**

window $T_v$ of consecutive lower-half frames and a speech segment $S$. After passing through the speech encoder and mask encoder, 512-dim embeddings $s = E_{\text{speech}}(S)$ and $m = E_{\text{mask}}(T_v)$ are obtained respectively. Cosine similarity distance and binary cross entropy loss are then calculated between the embeddings. The losses are formally defined as:

$$P_{\text{sync}} = \frac{s \cdot m}{\max(\|s\|_2 \cdot \|m\|_2, \epsilon)} \quad (3)$$

$$\mathcal{L}_{\text{sync}} = \frac{1}{N} \sum_i^N -\log(P_{\text{sync}}^i) \quad (4)$$

where $P_{\text{sync}}$ is a single value between $[0, 1]$ and $N$ is the batch size. $\epsilon$ is used to prevent division by zero. We train the lip-sync expert on the HDFT dataset [53] with a batch size of 8, $T_v = 5$ frames, $S = 0.2s$ segment, using the Adam optimizer with a learning rate of 1e-4. After approximately one day of training, the model converges. Our expert network eventually achieves 81% accuracy on the test set.

### 3.3 Segmentation-guided GAN Injection

The second sub-network illustrated in fig. 2 is the segmentation-guided GAN injection (SGI) network. Giving a portrait and its corresponding mask, SGI first encodes the image into the latent space to obtain the latent code, then inverts the generated latent code back to the image domain through style injection.

There exist various latent spaces such as $\mathcal{W}$, $\mathcal{W}+$ and $\mathcal{S}$ space. Many works [1, 25, 32, 38, 45] have investigated their representational abilities from the perspectives of distortion, perception, and editability. Here, we choose $\mathcal{W}^{c+}$ space, a variation of $\mathcal{W}+$ space originated from [19] as representation of latent code. To leverage this representation, a powerful encoder is required to accurately map each input image to a corresponding code. Although many encoders [25, 38, 45] have been proposed, they focus on extracting global latent code for global editing, such as age, emotion, making them unsuitable for textures disentanglement and local editing. To this end, we adopt a variation of [25] for latent code extraction. The encoder utilizes a feature pyramid network (FPN) [18] for feature fusion, ultimately generating fine-grained, medium-grained, and coarse-grained feature maps at three different scales. The mask is then resized to match each feature map. Subsequently, a global average pooling (GAP) is employed to extract semantic region features according to the segmentation, resulting in multi-scale style vectors. These are concatenated and further passed through an MLP to obtain the $\mathcal{W}^{c+}$ style codes.

Table 1: Quantitative comparisons on the HDTF dataset. The best performance is highlighted in red (1st best) and blue (2nd best). Facial Editing and Swapping Background demonstrate the method's capabilities for facial attribute editing and swapping background. For better visualization, we scale up the FVD by a factor of 0.1. Our SegTalker obtains the best visual quality, temporal consistency and comparable performance in lip synchronization. our method is the only approach that can simultaneously achieve facial editing and background swapping.

| Method | Local Editing | Swapping Background | Synchronization | | | Visual Quality | | | | |
|---|---|---|---|---|---|---|---|---|---|---|
| | | | Sync↑ | F-LMD↓ | M-LMD↓ | FID↓ | LPIPS↓ | PSNR↑ | SSIM↑ | FVD↓ × 0.1 |
| Real Video | N/A | N/A | 7.305 | 0 | 0 | 2.623 | 0.0109 | - | 1.000 | 1.048 |
| Wav2Lip [24] | N/A | N/A | **8.413** | 3.727 | 4.528 | **16.299** | 0.1031 | **31.821** | 0.921 | 17.423 |
| SadTalker [52] | Blink Only | N/A | 6.725 | 34.970 | 36.172 | 59.059 | 0.7567 | 15.967 | 0.496 | 85.638 |
| DiffTalk [31] | N/A | N/A | 4.615 | **2.004** | **1.614** | 23.498 | 0.111 | 31.654 | 0.913 | **16.606** |
| StyleHEAT [48] | Blink Only | N/A | 6.296 | 29.672 | 31.390 | 111.229 | 0.7969 | 14.968 | 0.465 | 91.2356 |
| AD-NeRF [10] | N/A | **Yes** | 5.216 | 3.574 | 3.825 | 18.614 | **0.1013** | 30.640 | **0.923** | 18.438 |
| Ours | **Yes** | **Yes** | **6.872** | **3.405** | **3.173** | **10.348** | **0.0494** | **33.590** | **0.934** | **9.205** |

Specifically, given a source image $I$ and its corresponding mask $M$, we first utilize a multi-scale encoder $E_\phi$ to obtain the feature maps $F = [F_i]_{i=1}^N$ at different resolutions:

$$F = E_\phi(I) \qquad (5)$$

Here $N$ is equal to three. We then aggregate per-region features based on the mask $M$ and features $F$. Specifically, for each feature map $F_i$, we first downsample the mask to match the feature map size, then perform global average pooling (GAP) to aggregate features for different regions:

$$u_{ij} = \text{GAP}(F_i \circ (\text{Down}(M)_i = j)), \{j = 1, 2, ..., C\} \qquad (6)$$

Where $u_{ij}$ denotes the averaged feature of region $j$ in feature map $i$, $C$ is the number of semantic regions, $\text{Down}(\ldots)$ is the downsampling operation to align with $F_i$, and $\langle \circ \rangle$ is the element-wise product. Subsequently, the multi-scale feature vectors $\{u_{ij}\}_{i=1}^N$ of region $j$ are concatenated and passed through a multi-layer perceptron (MLP) to obtain the style codes:

$$s_j = \text{MLP}([u_{ij}]_{i=1}^N) \qquad (7)$$

where $s_j$ denotes the style code of $j$-th categories. Then, the mask and style codes $s \in \mathbb{R}^{C \times 18 \times 512}$ are fed into the mask-guided Style-GAN generator to synthesize the talking face. For the detailed architecture of the Mask-guided StyleGAN, please refer to the supplement.

**Prior Learning** In order to seamlessly integrate SGI into the overall framework, we randomly select a mask from the images within a 15-frame range of the input image. Through such a training strategy, the model can learn the priors of semantic regions like teeth and eyes. Specifically, when given an image with closed mouth but a randomly selected mask corresponds to a visible-teeth state, it learns to model the teeth prior information and can naturally connect with the TSG module.

**Loss Functions** SGI is trained with a series of weighted objectives. Firstly, for the input image $I$ and generation image $\hat{I}$, we utilize the pixel-wise $\mathcal{L}_2$ and LPIPS [51] loss to have a better perceptual quality:

$$\mathcal{L}_2 = \| I - \hat{I} \|_2^2 \qquad (8)$$

$$\mathcal{L}_{\text{lpips}} = \sum_s \| V(I_s) - V(\hat{I}_s) \|_2^2 \qquad (9)$$

where $V(\ldots)$ denotes the perceptual feature extractor.

To prevent identity drift, we extract features from a pre-trained face recognition network [8] and maximize the cosine similarity between the input and generated images:

$$\mathcal{L}_{\text{id}} = 1 - \langle R(I), R(\hat{I}) \rangle \qquad (10)$$

where $R(\ldots)$ is the ArcFace model [8] and $\langle \ldots \rangle$ denotes the cosine similarity distance.

Moreover, a face parsing loss is also utilized following the work:

$$\mathcal{L}_{\text{parsing}} = 1 - \langle P(I), P(\hat{I}) \rangle \qquad (11)$$

where $P$ is the pre-trained face parser used in [17].

Using the losses mentioned above solely can not produce photorealistic results. Hence, we additionally employ an adversarial loss to enhance image quality, which is defined as:

$$\mathcal{L}_{\text{adv}} = \mathbb{E}[1 - \log D(\hat{I})] + \mathbb{E}[\log D(I)] \qquad (12)$$

where $D$ is initialized with the pre-trained StyleGAN discriminator. Finally, the overall objective function is summarized as:

$$\mathcal{L}_{\text{total}} = \mathcal{L}_2 + \lambda_{\text{lpips}} \mathcal{L}_{\text{lpips}} + \lambda_{\text{id}} \mathcal{L}_{\text{id}} + \lambda_{\text{parsing}} \mathcal{L}_{\text{parsing}} + \lambda_{\text{adv}} \mathcal{L}_{\text{adv}} \qquad (13)$$

where $\lambda_{\text{lpips}}$, $\lambda_{\text{id}}$, $\lambda_{\text{parsing}}$ and $\lambda_{\text{adv}}$ are the trade-off hyperparameters and set to 0.8, 0.1, 0.1, and 0.01, respectively.

## 4 EXPERIMENTS

### 4.1 Experimental settings

**Dataset** Since StyleGAN [13] typically generates high resolution images, e.g. 512 or 1024, while most existing talking face datasets have a lower resolution of 256 or below, we opt to train on the HDTF dataset [53] for high-quality talking face synthesis. The HDTF dataset is collected from YouTube website published in the last two years, comprising around 16 hours of videos ranging from 720P to 1080P resolution. It contains over 300 subjects and 10k distinct sentences. We collected a total of 392 videos, with 347 used for training and the remaining 45 for testing. The test set comprises videos with complex backgrounds and rich textures, thereby offering a comprehensive evaluation of the model performance.

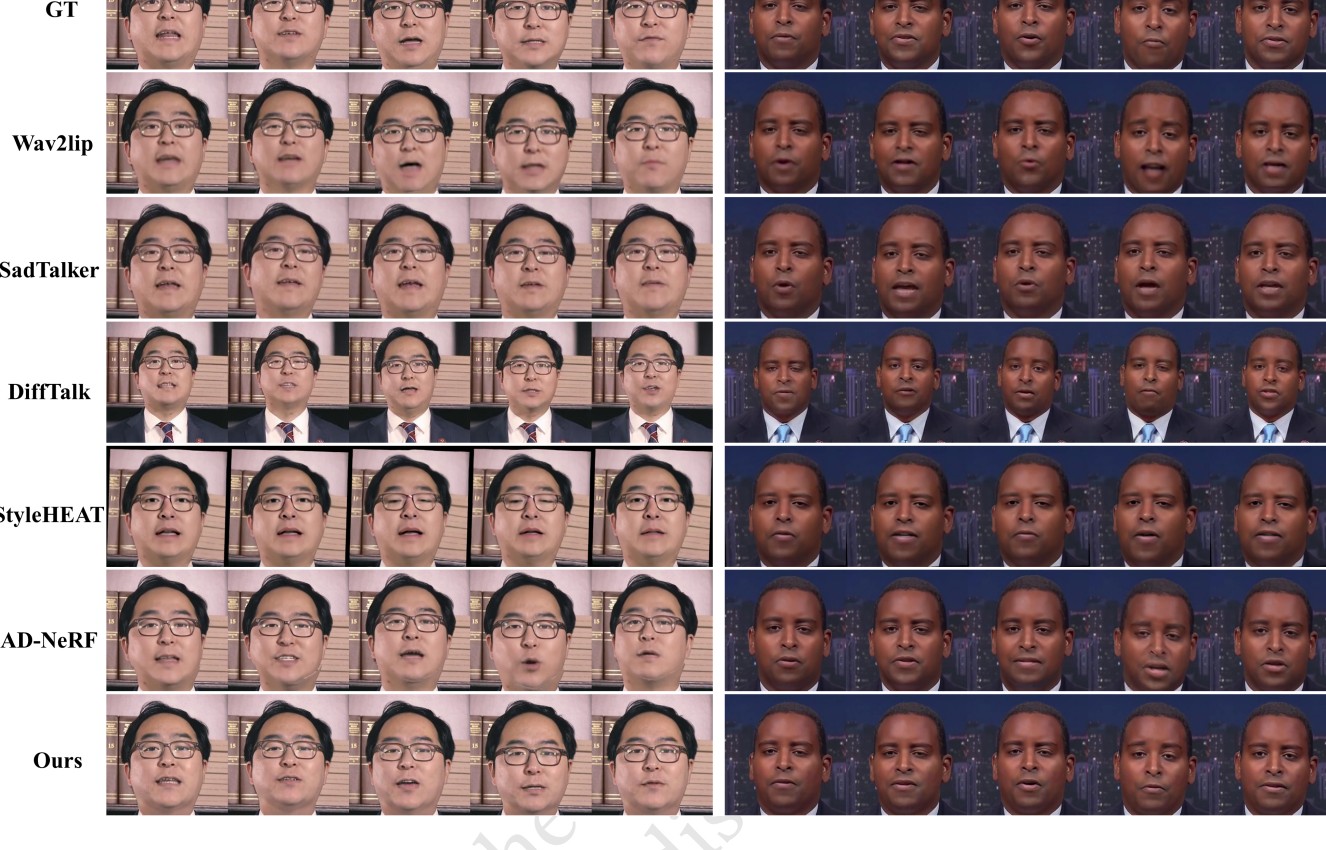

**Figure 3: Qualitative comparisons of our results with several state-of-the-art methods for talking face synthesis. our method produces high-fidelity video frames with rich textural details, while other methods struggle to preserve identity and contain artifacts. It is worth noting that AD-NeRF needs to train on these two identities respectively to produce the results.**

**Metrics** We conduct quantitative evaluations on several widely used metrics. To evaluate the lip synchronization, we adopt the confidence score of SyncNet [7] (**Sync**) and Landmark Distance around mouths (**M-LMD**) [5]. To evaluate the accuracy of generated facial expressions, we adopt the Landmark Distance on the whole face (**F-LMD**). To evaluate the quality of generated talking face videos, we adopt **PSNR** [41], **SSIM** [43], **FID** [11] and **LPIPS** [51]. To measure the Temporal coherence of generated videos, we employ **FVD** [39]. Higher scores indicate better performance for Sync, PSNR, and SSIM, while lower scores are better for F-LMD, M-LMD, FID, LPIPS and FVD.

**Implementation Details** We use PyTorch [23] to implement our framework. We train TSG module on a single NVIDIA A100 GPU with 40GB, while SGI module is trained on 4 NVIDIA A100 GPUs. In stage 1, We crop and resize face to 512×512. Speech waveforms are pre-processed to mel-spectrogram with hop and window lengths, and mel bins are 12.5ms, 50ms, and 80. The batch size is set to 20 and the Adam solver with an initial learning rate of 1e-4 ($\beta_1 = 0.5, \beta_2 = 0.999$) is utilized for optimization. In stage 2, we set the batch size to 4 for each GPU and initialize the learning rate as 1e-4 with the Adam optimizer ($\beta_1 = 0.9, \beta_2 = 0.999$). The generator

is initialized with StyleGAN weights [13]. In the first stage, we train the model with only the cross entropy loss for approximately 50K iterations, then incorporate the expert SyncNet to supervise lip movements for an extra 50k iterations. In the second stage, we train the model for 400K iterations.

## 4.2 Experimental Results

**Qualitative Talking Segmentation Results** In the first sub-network, we visualize the talking segmentation results illustrated in fig. 4. It can be observed that the generated segmentations effectively delineate distinct facial regions, even elaborating details such as earrings. Additionally, the synthesized lips exhibit strong synchronization with the ground truth. Subsequently, the high-quality segmentations produced by TSG are utilized as guidance of the SGI to deliver the final output.

**Quantitative Results** We compare several state of the art methods: Wav2Lip [24], SadTalker [52] (3DMM-based), DiffTalk [31] (diffusion-based), StyleHEAT [48] (styleGAN-based) and AD-NeRF [10] (NeRF-based). We conduct the experiments in the self-driven setting on the test set, where the videos are not seen during training. In these methods, the head poses of Wav2Lip, DiffTalk, and

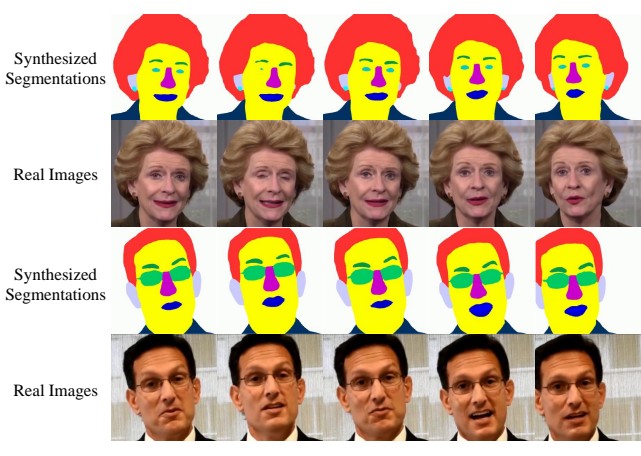

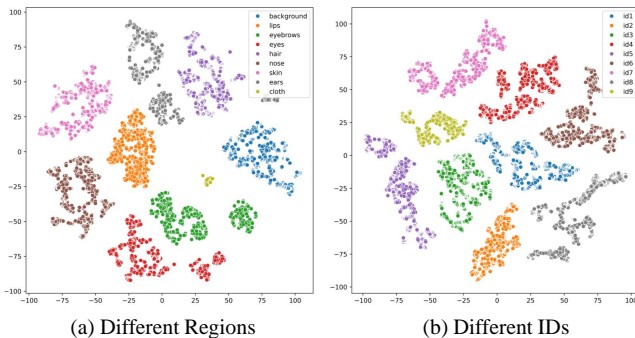

**Figure 4: Visualization of synthesized segmentation(row 1, row 2) and real images(row 2, row 4).**

(a) Different Regions    (b) Different IDs

**Figure 5: (a) Visualization of the regional style codes of a speaker. (b) Visualization of the style codes of 8 speakers in a particular region (here hair for example).**

SegTalker are fixed in their samples. For other methods, head poses are randomly generated. The results of the quantitative evaluation are reported in table 1.

Our method achieves better visual quality, temporal consistency, and also shows comparable performance in terms of lip synchronization metrics. Since DiffTalk takes ground truth landmark as conditional input, it is reasonable for DiffTalk to achieve the lowest LMD in self-driven sets. However, DiffTalk performs poorly in frame-to-frame coherence, especially with significant jitter in the mouth region (see supplementary video). In synchronization, despite scoring slightly lower on metrics relative to Wav2lip, our method achieves a similar score with ground truth videos. Furthermore, Our method outperforms existing state-of-the-art approaches on both pixel-level metrics such as PSNR, as well as high-level perceptual metrics including FID and LPIPS, thereby achieving enhanced visual quality. We additionally measure the FVD metric and Our FVD score is the best. This means that our method is able to generate temporal consistency and visual-satisfied videos. This is largely attributed to the implementation of SGI module. By explicitly disentangling different semantic regions via segmentation, SGI can better preserve texture details during image reconstruction.

Ref.    Source    Local Editing Results

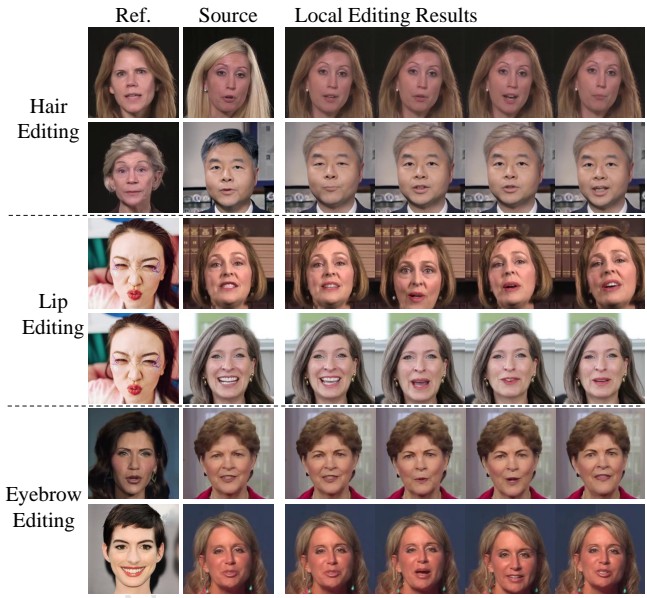

Hair Editing

Lip Editing

Eyebrow Editing

**Figure 6: Qualitative results of local editing. Our method produces more high-fidelity results in editing regions while maintaining the details and identity information of other regions.**

Moreover, our method is the only approach that can simultaneously achieve facial editing and background replacement which will be discussed in the following section.

**Qualitative Results** To qualitatively evaluate the different methods, we perform uniformly sampled images from two synthesized talking face videos which are shown in fig. 3. Specifically, the ground truth videos are provided in the first row where synthesized images of different methods follow the next and ours are illustrated in the bottom row. In comparison to Wav2lip [24], our results exhibit enhanced detail in the lip and teeth regions. For SadTalker [52], It employs single-frame animation, which inevitably causes background movement and generates artifacts when wrapping motion sequences. Additionally, it also cannot handle the scenarios with changing background. The incorporation of segmentation in our approach allows high-quality background replacement. DiffTalk [6] can generate visually satisfying results; however, diffusion-based methods still face significant challenges in terms of temporal consistency. The mouth area of DiffTalk is prone to shaking and leads to poor lip synchronization performance. StyleHEAT [48] is also a StyleGAN-based approach, but it cannot directly drive speech to generate talking face video. Instead, it requires the assistance of SadTalker to extract features from the first stage, then warps the features to generate video. Therefore, the quality of the video generated by StyleHEAT is limited by the quality of the output generated by SadTalker. AD-NeRF [10] is a NeRF-based method capable of generating high-quality head part but consistently exists artifact in the connection between the head and neck. Moreover, its inference is time-consuming (10s per image) and requires fine-tuning for each speaker (about 20 hours). In contrast, our method can produce more realistic and high-fidelity results while achieving

**Table 2: Quantitative comparisons of our methods under different ablation configurations. P.L. and C.S. denote prior learning and cross entropy, respectively.**

| Conf. | Sync↑ | FID↓ | PSNR↑ | SSIM↑ |
|---|---|---|---|---|
| w/o P.L. | 5.314 | 28.254 | 28.685 | 0.847 |
| w/o C.S. | 3.126 | 63.264 | 14.257 | 0.479 |
| w/o SyncNet | 4.174 | 10.647 | 32.036 | 0.913 |
| All pipeline | **6.872** | **10.348** | **33.590** | **0.934** |

accurate lip sync, satisfactory identity preservation and rich facial textures. For more comparison results, please refer to our demo videos in the supplement materials.

**Disentangled Semantics Visualization** To demonstrate the Disentanglement of the model across different semantic regions, we employ t-distributed stochastic neighbor embedding(t-SNE) [40] visualization to illustrate the per-region features, as depicted in fig. 5(a). For Clarity, We select eight sufficiently representative regions (appear in all videos) and utilize the mask-guided encoder to extract style codes from these semantic regions. In fig. 5(a), each region is marked with a distinct color. As shown, the style codes of same region cluster in the style space and different semantic regions are explicitly separated substantially. This demonstrates that our mask-guided encoder can accurately disentangle different region features. Furthermore, in fig. 5(b), we visualize the features of different IDs within a particular region to demonstrate the capability of the encoder. It can be seen that the style codes of different IDs are fully disentangled and our model can learn meaningful features.

**Facial Editing and Swapping Results** Our method also supports facial editing and background swaps while generating video. Given a reference image and a sequence of source images, our method can transfer the candidate region texture to the source images. As depicted in fig. 6, we illustrate three local editing tasks, including fine-grained hair editing, lip makeup, and eyebrow modifications. Besides, we also can manipulate blinking in a controllable manner by simply editing the eye regions of mask, illustrated in fig. 1. Compared with existing blink methods, our method does not design a specialized module for blinking editing, as well as enables other types of local editing, substantially enhancing model applicability and scalability. Additionally, our model intrinsically disentangles the foreground and background, allowing for seamless background swapping and widening the application scenarios of talking faces. As shown in fig. 7, with a provided reference background image and a video segment, we can not only generate synchronized talking face video, but also achieve video background swapping, resulting in high-fidelity and photo-realistic video.

### 4.3 Ablation Study

In this section, we perform an ablation study to evaluate the 2 sub-agent networks, which are shown in table 2. We develop 3 variants with the modification of the framework corresponding to the 2 sub-network: 1) *w/o prior learning*, 2) *w/o cross entropy* and 3) *w/o SyncNet*.

**The first component** is the implementation of *priors learning*. Without *priors learning*, the method produces poor visual quality.

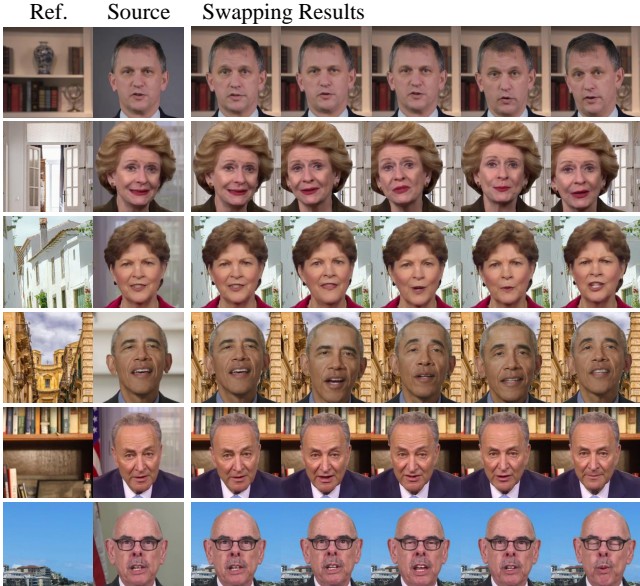

Ref.   Source   Swapping Results

**Figure 7: Example of Swapping background. Given a video and a background image, our method can produce natural and photo-realistic swapping videos.**

This mechanism offers structural prior information for the mouth and teeth regions, which facilitates the model learning personalized details of these areas. **The second component** is the *cross entropy*. Without *cross entropy*, the method exhibits very poor performance whenever on both lip synchronization and visual quality. By employing cross-entropy loss instead of L1 loss, we overcome the issue of erroneous segmentation predictions around region boundaries, improving the model's control over different semantic areas. Furthermore, *cross entropy* also facilitates learning lip movements from speech, exhibiting a certain extent of lip synchronization. **The last component** is the *SyncNet*, which is performed to reinforce the model learning the mapping from speech to lip. The performance of visual quality is comparable to the baseline when we do not apply *SyncNet*. However, without *SyncNet* would lead to poor lip synchronization performance, which are demonstrated in table 2.

## 5 CONCLUSION

In this paper, we present a new framework SegTalker for talking face generation, which disentangles the lip movements and textures of different facial components by employing a new intermediate representation of segmentation. The overall framework consists of two sub-networks: TSG and SGI network. TSG is responsible for the mapping from speech to lip movement in the segmentation domain and SGI employs a multi-scale encoder to project source image into per-region style codes. Then, a mask-guided generator integrates the style codes and synthesized segmentation to obtain the final frame. Moreover, By simply manipulating different semantic regions of segmentation or swapping the different textures from reference image, Our method can seamlessly integrate local editing and support coherent swapping background.

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
