# OpenReview forum: "SegTalker: Segmentation-based Talking Face Generation with Mask-guided Local Editing"
_acmmm.org/ACMMM/2024/Conference — MM2024 Poster_

### Official Review · Reviewer_iXHb · 2024-05-20

**Rating:** 4
**Confidence:** 3

**Summary:**

This work introduces SegTalker, a novel architecture to decouple lip movements and image textures by introducing segmentation as intermediate representation. A mask-guided StyleGAN is also proposed to synthesize video frames. This work can accomplish a variety of tasks such as generating talking faces, swapping background, and local editing.

**Strengths:**

1.	The whole pipeline is technically sound and clear. In particular, Segmentation-guided GAN Injection fully combines stylegan's generative priors and the region localization of segmentation models.
2.	The authors conducted extensive comparative experiments and ablation studies to validate their method.
3.	This work can accomplish a variety of tasks and expand the application range of the model.

**Limitations:**

1.	Bisenet and Stylegan models need the input images to meet the face alignment of FFHQ, so the input with other alignment modes may not be well synthesized. Based on this, I would like to know the specific implementation method of the video results in the supplementary materials 00:18~01:08.
2.	This work only conducts training on HDTF dataset, and lacks further demonstration on VFHQ/VoxCeleb and other datasets, especially the use of additional reference identities for cross-id driving experiments.
3.	Similar to the reasons in 1, for the fairness of comparison, I suggest the author to re-check whether all models in Table 1 use the same facial alignment and cropping methods. Based on the results of DiffTalk and the other methods shown in Figure 3, it does not appear to be a uniform approach to data processing.
4. Two questions need to be further explained about the local editing results shown in Figure 6. The first question is what causes hair editing in the second line to change the size of the hair area. Second, the eyebrow editing result presented seemed not to have been well edited. Does it mean that the method in this paper has a poor editing effect on fine-grained areas？

**Suitability:**

3

---

### Official Review · Reviewer_JeEF · 2024-05-22

**Rating:** 2
**Confidence:** 4

**Summary:**

The paper presents a framework for audio-driven talking face generation. The proposed method, SegTalker, decouples lip movements and image textures using segmentation as an intermediate representation. This framework involves two main components: the Talking Segmentation Generation (TSG) module, which synchronizes lip movements with input audio, and the Segmentation-guided GAN Injection (SGI) network, which synthesizes high-quality video frames. The approach not only ensures temporal consistency and high visual quality but also allows for localized facial editing and background replacement, enhancing the flexibility and application scope of the generated videos.

**Strengths:**

- The paper introduces a unique approach by employing segmentation to disentangle lip movements from textural details.
- The framework has the ability to perform local facial editing and background swapping which is useful for several applications.
- The paper provides comprehensive quantitative and qualitative evaluations with several state of the art models.

**Limitations:**

- The paper evaluates only on a single dataset HDTF, where most of the current talking head generation papers are evaluated on more than one.
- The paper does not mention the generalisation of the model to unseen identities or speech audio. It also does not state if the test set contains speakers from the training set or not.
- No user study conducted in the paper.


- The paper only mentions StyleHeat as StyleGAN-based talking head generation model, where there are other works that use the StyleGAN model to generate talking head videos such as "Style2Talker: High-Resolution Talking Head Generation with Emotion Style and Art Style" by Tan et al published at AAAI 24 and "Talking head from speech audio using a pre-trained image generator" by Alghamdi et al, published at ACM MM 22.

**Suitability:**

3

---

### Official Review · Reviewer_EuXr · 2024-05-25

**Rating:** 4
**Confidence:** 3

**Summary:**

The paper introduces SegTalker, a novel framework designed to enhance audio-driven talking face generation by leveraging segmentation as an intermediary representation to decouple lip movements from image textures.

**Strengths:**

1.SegTalker excels in producing videos with lip movements that are both highly synchronized to the input audio and rich in detail, particularly regarding skin and teeth textures, enhancing visual authenticity.
2.Unique to SegTalker is its capacity for background swapping and detailed local editing, such as blinking, while preserving the integrity and quality of other facial regions, offering a versatile editing toolset.
3.By employing segmentation as a guiding principle during image reconstruction, SegTalker effectively retains fine-grained details in textures, leading to more visually pleasing results when modifying or generating talking faces.
4.The method unifies talking face synthesis with controlled editing within a single, end-to-end framework, streamlining the process and ensuring coherent outputs across different manipulations.

**Limitations:**

1. Although specific computational requirements are not detailed, the multi-step process involving speech-driven segmentation generation, segmentation-guided encoding, and StyleGAN synthesis suggests that SegTalker may demand significant computational resources and processing time.
2. The performance of SegTalker partly hinges on the quality and versatility of the pre-trained networks used for segmentation, which could limit its applicability across different datasets or necessitate fine-tuning for specialized scenarios.
3. While qualitative and quantitative assessments are provided, the comparison with a limited set of baseline methods does not comprehensively benchmark SegTalker against the latest advancements, potentially underrepresenting its relative strengths.

**Suitability:**

3

---

### Official Review · Reviewer_zA7n · 2024-05-26

**Rating:** 2
**Confidence:** 3

**Summary:**

The paper aims to generate talking-face videos by preserving facial textures (like skin and teeth) which are not directly addressed in the existing works. The framework also enables local facial editing like editing or swapping hair, eyebrows etc. The evaluation is done on a small-scale high-quality public dataset. However, no specific evaluation or comparison has been shown regarding the facial textures. Also,  the lip-synchronization quality is considerably poorer and seems practically unusable.

**Strengths:**

- The ability of the proposed network to perform facial editing along with talking-face video generation is impressive. The facial editing results also look reasonably good.
- Facial editing results are string and convincing.

**Limitations:**

- Lip-sync quality: From the video results in supplementary, it is very clear that the quality of lip-sync is poor. In many instances, either the lip shape is not generated accurately, or the lips are out-of-sync. This is also reflected in the main results (Table-1). This poses a huge risk to the proposed method since without accurate lip movements, the overall quality and feel of the generated talking-face video reduces significantly.
- Evaluation for textures: One of the major motivation and the claim of the paper is that the proposed method preserves the facial textures (like hair and teeth) which are not addressed in previous works. However, I could not find any evaluation to support this claim. Even the qualitative results in the supplementary do not particularly compare and analyse the texture aspect. Without convincing evaluation, it is hard to see the benefits of the proposed approach.
- Need for mask-based approach: I am not convinced why is the mask-based approach better suited for talking-face generation task, specifically when it comes to generating lip movements. A lot of fine-grained facial information can be lost in generating the lip motion on a masked face.
- Evaluation on more datasets: The current method is trained and evaluated on a small-scale dataset of ~16 hours. This makes it challenging to fully understand the capabilities of the proposed framework. The authors can consider using other publicly available high-quality talking-face video datasets like MEAD [1] to validate the robustness of the network.
- Paper writing: The paper is not very easy to follow. The need for multiple components, and the structure of the paper could be re-organised to make it more easy to comprehend.

[1] Kaisiyuan Wang et al. MEAD: A Large-scale Audio-visual Dataset for Emotional Talking-face Generation. In ECCV 2020.

**Suitability:**

3

---

### Meta-Review · Area_Chair_hMD9 · 2024-06-30

**Recommendation:** Accept (Poster)
**Confidence:** 3

**Metareview:**

The paper jointly addresses the problems of speech-driven talking head generation and local face editing guided by segmentations masks.

This paper is critical as it has mixed evaluations, with some reviewers (BA, BA) that did not provide a final rating after rebuttal. One reviewer has recommended WA while the final justification suggests they wanted to recommend WR. This is not clear. They asked if they could evaluate the new results, which might have positively affected their rating.

Overall, the main positive aspects are:
- The problem formulation is new as it seems that previous methods perform either one of the two tasks separately, making it an interesting and novel framework.
- Facial editing results are convincing and the generated videos outperform previous methods.
- Despite a little degradation in lip-sync, the generated videos look very realistic and detailed in texture if compared with previous solutions.

Main weaknesses:
- Evaluation on one dataset only. In the rebuttal, authors provided results on an additional dataset, which are consistent with that reported in the main paper, suggesting the results in the paper are general.
- Some recent methods have not been considered in the experiments.
- The need of a segmentation map might impair results if the mask is not properly detected.

Regarding the two reviewers missing the final recommendation, the AC believes that the rebuttal was convincing in clarifying their concerns. In fact, most metrics refer to image/texture quality (SSIM, PSNR etc); further, the experimental setup and the face parser used look consistent and fair.

Overall, the reviewers scores seems in favor of the paper acceptance. Given that some final ratings are not given, the AC would tend to positively consider their first assessment, and positively evaluate the rebuttal. Ultimately, the AC preliminary rating would be accepting the paper. The proposed framework, even if not uniquely outperforming previous methods in terms of all metrics, provides a quite different alternative that looks worth presenting.